# Study on the Quantitative Evaluation of the Surface Force Using a Scanning Probe Microscope

**Wataru Yagi [1],\*, Tomomi Honda [1], Kazushi Tamura [2] and Keiichi Narita [2]**

[1]  Department of Mechanical Engineering, University of Fukui, 3-9-1 Bunkyo, Fukui-shi, Fukui 910-8507, Japan; honda@u-fukui.ac.jp

[2]  Idemitsu Kosan Co., Ltd. Lubricants Research Laboratory, 24-4 Anesakikaigan, Ichihara-shi, Chiba 299-0107, Japan; kazushi.tamura.0220@idemitsu.com (K.T.); keiichi.narita.0440@idemitsu.com (K.N.)

\*  Correspondence: jm190114@u-fukui.ac.jp

**Abstract:** There are two types of friction modifiers (FMs) used as lubricant additives: Reaction film FMs (RF-FMs) and adsorption film FMs (AF-FMs). While RF-FMs provide good performance in severe conditions, AF-FMs excel in mild conditions. This empirical evidence leads us to combine these two FMs to cover broader conditions. However, the effects of their combination are highly complicated due to the interaction between these FMs. If the interaction force of AF-FMs with various materials can be evaluated, it would help us to improve tribological performances of lubricants. Although atomic force microscopy seems suitable for this application, we found some obstacles, such as fluid resistance, electrostatic force, and laser positioning of the cantilever, to achieve proper measurements of the adsorption force. In this study, the adsorption force between the polar group and the surface was directly measured in oil with a 1 μm silica probe modified with $CH_3$ or COOH. This paper proposed how to eliminate errors included in the adsorption force measurement using AFM and a calibration method for obtaining an accurate adsorption force of the polar group, and a test of normality of the measured data was conducted by 400 measurements. As a result, it was shown that approximately 100 tests were needed to obtain an accurate adsorption force in this study.

**Keywords:** friction modifiers (FMs); adsorption force measurement; atomic force microscopy (AFM); colloid probe; calibration

## 1. Introduction

The automobile industry is required to produce energy-saving vehicles. Holmberg et al. [1] reported that the breakdown of the energy conversion rate of an automobile engine is as follows: Energy to move the car, 21.5%; friction loss, 16.5%; and wasted energy losses (e.g., from exhaust and cooling systems), 62%. In addition, they have reported that friction reduction contributes considerably to energy savings because it also reduces losses by the exhaust and cooling systems. Low-viscosity oil is effective for friction reduction. However, low-viscosity oil is less effective for lubrication, and it also cannot suppress the wear of sliding surfaces. Thus, lubricant additives are used to further improve lubrication. Oil additives include, for example, friction modifiers (FMs) and anti-wear agents. FMs are classified as either reaction film FMs (RF-FMs) or adsorption film FMs (AF-FMs). RF-FMs reduce friction by chemically reacting with the metal surface and forming a coating film [2,3]. Zinc dithiophosphate and molybdenum dithiocarbamate are classified as RF-FMs. By contrast, AF-FMs reduce friction by adsorbing onto the metal surface. Ester and amine are classified as AF-FMs and they are called ash-free FMs because they do not contain sulfur, phosphorus, or metallic elements. While RF-FMs provide good performance under severe conditions because they need energy for reacting and forming a coating film, AF-FMs excel under mild conditions because they can sustain the adsorption film. In addition,

the adsorption mode is classified as either chemisorption or physisorption. It is generally thought that the chemisorption force is larger than the physisorption force and a large chemisorption force gives high wear resistance. However, chemisorption causes corrosion and chemical wear. By contrast, the physisorption force cannot resist a large load, during which it cannot reduce friction, although it is not necessary to consider corrosion and chemical wear, because the physisorption mode is derived from interaction forces between atoms and molecules, such as van der Waals forces and electrostatic forces. This empirical evidence leads us to combine these FMs to cover broader conditions. However, the effects of their combination are highly complicated due to the interaction between these FMs. Guegan et al. [4] demonstrated that optimal combinations of FMs, which contain organic FMs (OFMs), organomolybdenum FMs (MoFMs), and first-polymer FMs, provide a friction coefficient lower than that achieved using only one of the FMs. However, some combinations of FMs, for example, the combination of MoFMs and OFMs, give higher coefficients than those obtained only using either FM because the OFMs slow the adsorption of MoFMs on the surface. In addition, Campen et al. [5] showed that oil temperature has an effect on the adsorption of FMs onto the surface and the resulting friction coefficient. By contrast, Hirano et al. [6] reported that the OFMs improve performance in terms of friction reduction, wear, and seizure when the hydrocarbon chain length is similar for both the OFMs and base oil. However, Jahanmir [7] found no such benefits from matched chain lengths. Thus, although high-performance ash-free FMs help us to develop a lubricant covering a wide range of sliding conditions, it is presently a significant challenge to develop the appropriate FMs for each sliding condition.

The properties of adsorption films are determined by the properties of the additive molecules, such as hydrocarbon chain length, structure, and adsorption mode onto the surface. These properties have been investigated by frequency modulation, including atomic force microscopy (FM-AFM), quartz crystal microbalancing (QCM), and neutron reflectometry (NR). Hirayama et al. [8] reported that cross-sectional images of the adsorption layer were acquired by FM-AFM to observe in situ the adsorption of fatty acids onto metal surfaces. In addition, they revealed that the adsorbed additive layer gradually grew to a thickness greater than about 20 nm due to an external stimulus, such as cantilever oscillation. FM-AFM enables us to directly observe the adsorption film structure and grasp whether the additive has an adsorption ability. Furthermore, Lundgren et al. [9] showed by QCM that oleic acid molecules appear to adsorb in an essentially flat configuration on a steel surface, but that linoleic acid molecules accumulate in a more extended form. Moreover, Hirayama et al. [10,11] demonstrated by NR that the adsorption film thickness of fatty acids has about the same monolayer thickness as that of single fatty acid molecules and the adsorption layer changes the monolayer to a multilayer when a pressure field is added. QCM and NR help us to determine the adsorption ability, quantity of molecules, and adsorption film thickness without physically contacting the film. However, the data obtained from these techniques reveal only the adsorption film thickness and whether the additive has adsorption ability, and do not provide information about the adsorption force. These are important data for the development of high-performance FMs, and the adsorption force between the polar group of an FM and the surface is also important data for the development of high-performance FMs.

The surface force apparatus (SFA) and AFM have been used to measure the interaction force directly. Zhu et al. [12] confirmed that saturated alkyl chain OFMs adsorb onto the surface and form vertically oriented monolayers on mica. Furthermore, they found that the FM molecules increased the distance between sliding surfaces and reduced the adhesion force and the magnitude of the shear force. The SFA enables us to measure the adhesion force between the surfaces. However, SFA also cannot directly measure the adsorption force between the polar groups of FMs and the surface. In addition, the SFA restricts the solid surface material to mica, while AFM does not. Moreover, AFM enables us to measure the interaction force between the sub-micrometer probe and the surface. The colloidal probe method, in which a colloidal particle with a chemically modified polar group is used as the probe, helps us to simulate a real system.

The adsorption force between the polar group and the surface has been directly measured in oil with a 1 μm silica probe modified with $CH_3$ or COOH in our laboratory [13]. In the present study, it was suggested that the force data contained some error due to various factors, such as fluid resistance, electrostatic force, and laser positioning of the cantilever. Ducker et al. [14] examined the adhesion force between the silica probe and the mica surface in an aqueous solution by changing the concentration and pH by AFM, and they concluded that the AFM-measured decay length of the electric double-layer is predicted well by the Derjaguin-Landau-Verwey-Overbeek (DLVO) theory. Thus, measurement of the decay provided a simple method to calibrate the expansion of AFM piezoelectric crystals perpendicular to the surface. However, the DLVO theory is applied to medium polarity with a relative dielectric constant of $\varepsilon_R > 5$, whereas a low-dielectric medium would fundamentally diverge from DLVO behavior [15]. For this reason, AFM measurement in a low-dielectric medium such as oil is not easy and it is still not clear what the error factors are and how the error should be processed. The present study revealed the error factors and proposed an error analysis method and appropriate processing to quantitatively evaluate the force and displacement at the nanoscale.

## 2. Experimental Approach

### 2.1. AFM Apparatus

All measurements were conducted in oil with AFM (SPM-9600, Shimadzu, Kyoto, Japan) using a small liquid cell and a cantilever (MLCT-O10, Bruker, Massachusetts, MA, US) suitable for AFM, and the cantilever was chemically modified by Novascan Technologies, Inc. (Product name: PT.GS.AU.CH$_3$ and PT.GS.AU.COOH, Novascan Technologies, Iowa, IA, US). Specifications of the cantilever are shown in Table 1. We examined the spring constant and confirmed it to be within the manufacturer's specification. Further, we assumed that the COOH group adsorbs more strongly than the $CH_3$ group onto a surface because of polarity differences. To obtain force curves in oil, a special cantilever holder with a glass slit at the top was used, and the oil was sandwiched between the glass slit and substrate. The AFM unit was placed on a controlled antivibration table. The room temperature and humidity were kept constant at 21 ± 1 °C and 60 ± 3%, respectively.

**Table 1.** Specifications of cantilever for AFM force measurement.

| | |
|---|---|
| Length, μm | 300 |
| Spring constant, N/m | 0.015 |
| Probe material | $SiO_2$ |
| Probe radius, μm | 1 |
| Probe surface | $CH_3$, COOH |

### 2.2. Substrate

Mica and silicon wafers were used as substrates. Kawai [16] demonstrated that the adhesion force between a probe tip and the substrate depends on the asperity of the substrate when the root-mean-square surface roughness $R_q$ is larger than 3 nm. Thus, in this study, it was confirmed that the surface roughness $R_q$ of the mica and silicon wafers was less than 3 nm, as measured by dynamic-mode AFM, and is shown in Table 2. Jones et al. [17] reported that the adhesion force is affected by the water film formed on the surface by the atmosphere. Therefore, the mica was cleaved in oil to avoid contact with the atmosphere. The silicon wafer (p-type, 110-oriented) was cleaned with ethanol in an ultrasonic bath.

**Table 2.** Root-mean-square surface roughness of the specimens.

| Specimens | Root Mean Square $R_q$, nm |
|---|---|
| Mica | 0.34 |
| Silicon wafer | 1.59 |

## 2.3. Oil

Paraffinic oil without additives was used as the sample oil because this oil is nonpolar and has stable molecules, like commonly used industrial base oils. Paraffinic oil is classified as Group II by the American Petroleum Institute. The specifics of paraffinic oil are as follows: Viscosity of 41.5 mm²/s at 23 °C and viscosity index of 116.

## 2.4. Force Mesurement

An example force curve is shown in Figure 1. The *y*-axis shows the voltage detected with a photodetector, which translates to the amount of deformation of the cantilever. The *x*-axis shows the vertical movement of the piezoelectric element. Curve (1) shows the process as the probe approaches the substrate. In (2), the substrate exerts a repulsive force on the probe, although this does not necessarily mean that the probe contacts the substrate. Curve (3) shows the process in which the probe is released by or recedes from the substrate. The photodetector voltage becomes negative when the restoring force of the cantilever becomes larger than the attractive force between the probe and substrate. The size of this negative bump in the curve represents the adhesion force $F_{ad}$ between the probe and substrate. In this paper, the *x*-axis was defined as 0 V, or the point of maximum repulsive force. The load on cantilever $P$ is determined by sweep distance $L$ and sweep start position $L_s$. When $L > L_s$, the cantilever is deformed in an upward direction by the difference between $L$ and $L_s$ after detecting a repulsive force. Thus, the maximum deflection distance was converted to the load of the cantilever with respect to substrate $P$ by multiplying it by the spring constant of the cantilever $k$. Sweep speed $v$ is determined by sweep distance $L$ and the frequency of piezoelectric force $f$, as shown in Equation (1).

$$v = 2L \times f \tag{1}$$

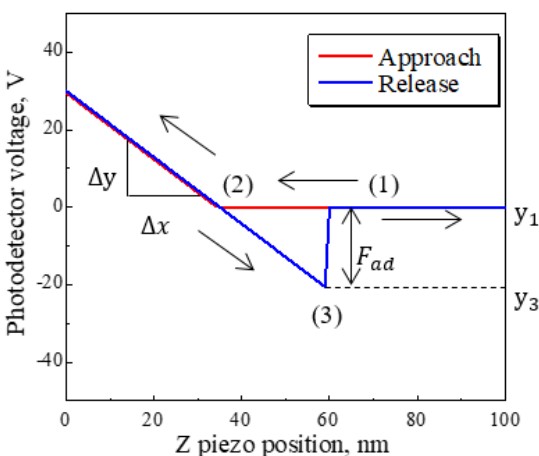

**Figure 1.** Example of a force curve measured by AFM. The *y*-axis reflects the deformation of the cantilever when it detects forces. The *x*-axis shows the vertical moving distance. (1) and its arrow indicate the approach process and force non-detection force area. (2) indicates the area where repulsive forces are detected. (3) indicates the process of release and retraction from the substrate point. $F_{ad}$ shows the adhesion force that is converted by Equation (2).

When the sweep distance $L$ or frequency $f$ become larger, the sweep speed $v$ increases. It is difficult to unify the load on cantilever $P$ with only sweep distance $L$ and sweep start position $L_s$ because point (2) in Figure 1 slips by Brownian motion, even at room temperature. Therefore, the load $P$ was unified

by changing the sweep start position $L_s$ to establish the sweep speed $v$. The adhesion force $F_{ad}$ between the probe and substrate was converted by Equation (2).

$$F_{ad} = k \times \frac{\Delta x}{\Delta y} \times (y_1 - y_3) \qquad (2)$$

The distance of the piezoelectric scanner per unit of voltage detected by the photodetector after the probe detects repulsive force $S$ (hereafter, the photodetector (PD) sensitivity) is needed to convert the voltage to an adhesion force. PD sensitivity is determined as shown in Equation (3).

$$S = \frac{\Delta x}{\Delta y} \qquad (3)$$

The difference between the values of Y at (1) and (3) in Figure 1 was converted to the adhesion force $F_{ad}$ by multiplying the PD sensitivity $S$ and cantilever spring constant $k$.

Force curves were corrected for the force non-detection area shown as (1) in Figure 1 because of the shift from the approach to the release process. The shift was eliminated by calculating a slope and y-intercept in the force non-detection area and subtracting the linear function from the raw force curve. In this study, all such values were converted to obtain corrected force curves.

## 3. Results and Discussion

### 3.1. Effect of Sweep Speed on Force Curve

Mica was used as a substrate. Force curves were obtained under the following conditions: Sweep distance $L$ = 3000 nm; sweep start position $L_s$ = 2500 nm; and sweep speed $v$ = 600, 1200, 1800, and 3000 nm/s. The force curves measured by the $CH_3$ and COOH probes are shown in Figure 2. In Figure 2, (a) and (c) show the approach process, and (b) and (d) show the release process. These force curves have significant differences. When the $CH_3$ probe was used, fluid resistance to the cantilever gradually increased according to increasing sweep speed. It was found that the cantilever encountered resistance in the fluid before detecting a repulsive force from the substrate. It is necessary to set the sweep speed as low as possible to measure the adhesion force exactly because the force curve is corrected by using the area of force non-detection. However, when the COOH probe was used, the probe detected a small attractive force and repulsive force in both the approach and release process.

Then, to examine the cause of these small forces, sweep distance $L$ and sweep start position $L_s$ were set as $L < L_s$, and we investigated the distance over which small forces are exerted on the probe. Force curves were obtained under the following conditions: Sweep distance $L$ = 1000 nm; sweep start positions $L_s$ = 1000, 1500, 2000, 2500, and 3000 nm; and sweep speed $v$ = 300 nm/s. The approach process with the COOH probe is shown in Figure 3. It was found that the probe began to detect the small forces at a distance of approximately 500 nm as the sweep starting position decreased.

The repulsive forces acting on the probe before it contacts the substrate are assumed to be (i) van der Waals force, (ii) steric repulsive force, and (iii) electrostatic double-layer force. The van der Waals force only acts at a distance within approximately 10 nm, and the steric repulsive force only acts at a smaller distance because it results from the Pauli exclusion principle. Therefore, these forces do not act at a distance of 500 nm. The electrostatic double-layer force results from the surfaces having a positive or negative charge. However, it was thought that the surfaces are not charged, because the oil has a low dielectric constant. Instead, we assumed that the small forces are electrostatic because the mica and oil are insulators. Thus, we conducted a test to evaluate the effect of the electrostatic force on the force curve.

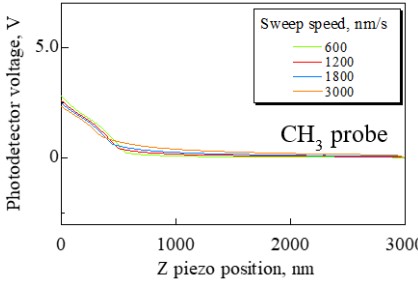

(**a**) Approach process between CH$_3$ probe and mica

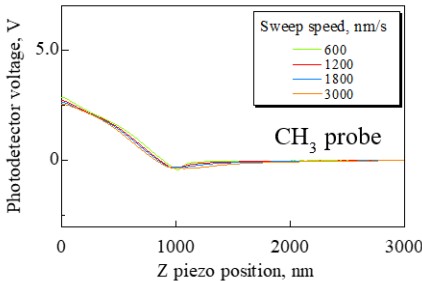

(**b**) Release process between CH$_3$ probe and mica

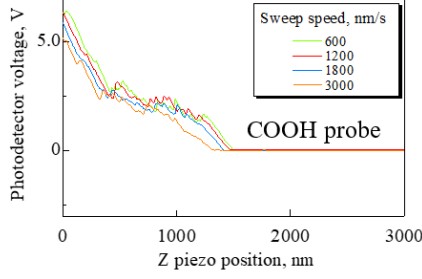

(**c**) Approach process between COOH probe and mica

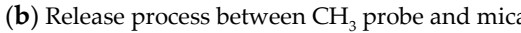
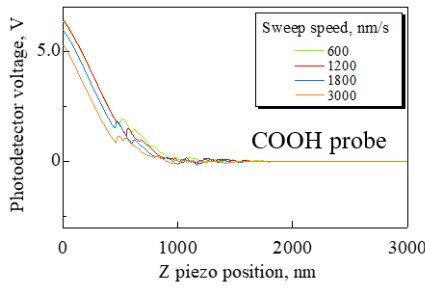

(**d**) Release process between COOH probe and mica

**Figure 2.** Approach processes and release processes using the CH$_3$ and COOH probes. (**a**) and (**c**) are the approach processes. (**b**) and (**d**) are the release processes. In (**a**), using the CH$_3$ probe, the faster sweep speed produces increased fluid resistance to the cantilever. In (**c**) and (**d**), the COOH probe is detecting small attractive and repulsive forces. It is suggested that the small forces result from the electrostatic force.

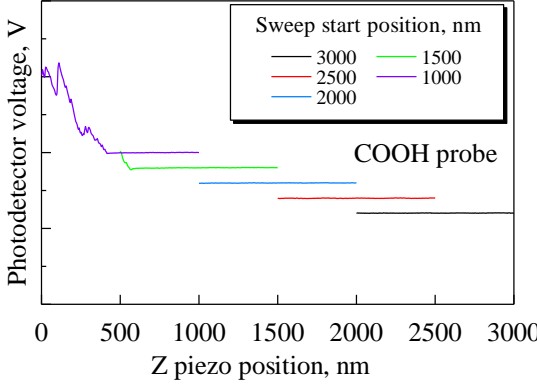

**Figure 3.** Approach processes when sweep distance $L$ is shorter than sweep start position $L_s$. For simplicity, each data set is displayed with a shift in the *y*-axis. For $L_s$ = 1500 and 1000 nm, the small forces are detected at approximately 500 nm. This distance indicates that the small forces are electrostatic.

### 3.2. Effect of Electrostatic Force on Force Curve

The method of attaching and securing the substrate was changed from a general double-sided tape to a carbon double-sided tape to examine the effect of the electrostatic force. The attachment method using general tape and a carbon tape is shown in Figure 4. The measured resistance of these tapes showed that, while the general tape is an insulator, the carbon tape is a conductor. Therefore, the carbon tape was expected to ground the electrostatic force. If the small attractive and repulsive forces result from electrostatic force, it is assumed that the small forces will decrease as time goes on due to charge relaxation. Then, force curves were obtained under the following conditions: Sweep

distance $L$ = 3000 nm, sweep start position $L_s$ = 2500 nm, sweep speed $v$ = 600 nm/s, and relaxation time of 0–60 min. The approach processes of these force curves are shown in Figure 5. It was found that the probe detects a long-range attractive force, although the probe does not detect the small forces without relaxation. In addition, the long-range attractive force gradually decreased over time, and the long-range attractive force disappeared after 60 min. Then, to compare the adhesion forces, the force curves without relaxation and after 60 min are shown in Figure 6. Figure 6a shows the force curve without relaxation with the COOH probe. Figure 6b shows the force curve after 60 min with the COOH probe. Figure 6c shows the force curve after 60 min with the CH$_3$ probe. Every plot also displays the adhesion force.

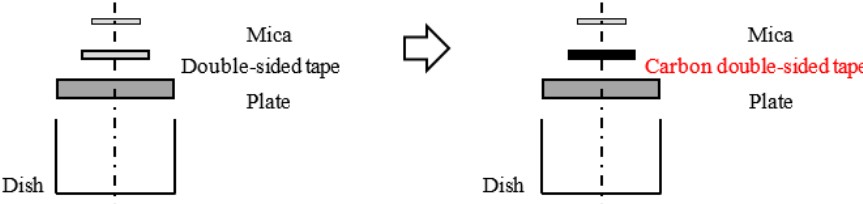

**Figure 4.** Effect of changing method of fixing from double-sided tape to carbon tape. Mica and oil are insulators. To ground mica with the tape, a carbon double-sided tape was substituted for a general double-sided tape to hold the mica. The resistance of the general tape and carbon tape shows that the general tape is an insulator and the carbon tape is a conductor.

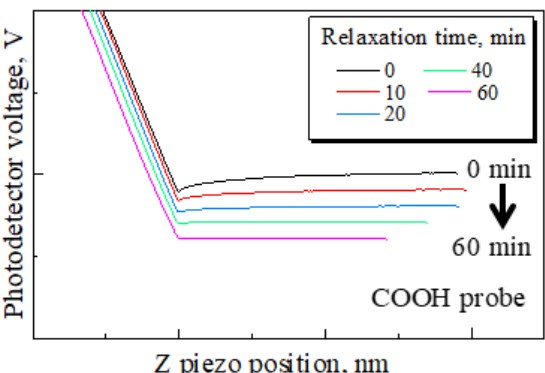

**Figure 5.** Changes in approach process by increased relaxation time. For simplicity, each data set is displayed with a shift in the *y*-axis. The small forces seen in Figure 3 disappear by substituting the general tape with carbon tape. However, a long-range attractive force is detected without relaxation time and after 10 and 20 min of relaxation. The long-range attractive force is eliminated by a relaxation time of 60 min.

These results demonstrated that almost all electrostatic forces are eliminated by using carbon tape to attach the substrate. However, a long-range attractive force appeared, and this force gradually decreased over time. Jones et al. [17] reported a similar attractive force in the atmosphere, and Barbagini et al. [18] reported the same force in *n*-dodecane. Both papers indicate that the long-range attractive force also results from electrostatic forces. Barbagini et al. showed that the long-range attractive force disappears in *n*-dodecane after about 60 min. Therefore, it is evident that the long-range attractive force in our study resulted from the electrostatic force. Then, the adhesion force after 60 min with the COOH probe was 3.3 nN. The adhesion force decreased by 50% compared to the adhesion force with no relaxation time. It is suggested that the decrease in the adhesion force is due to reduced electrostatic force, and 3.3 nN reflects the adsorption force of COOH. Hence, it is essential to attach the substrate with conductive tape and relieve charges by applying a relaxation time of about 60 min to measure exactly the adsorption force of the polar group.

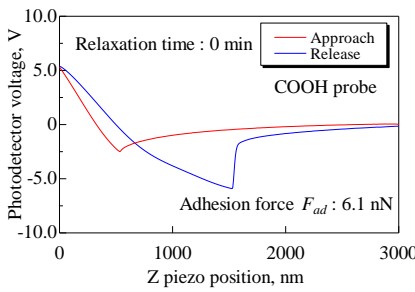

(**a**) No relaxation time force curve using COOH probe

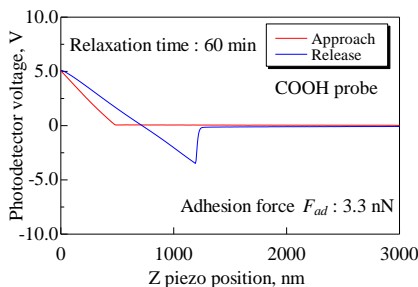

(**b**) After 60 min force curve using COOH probe

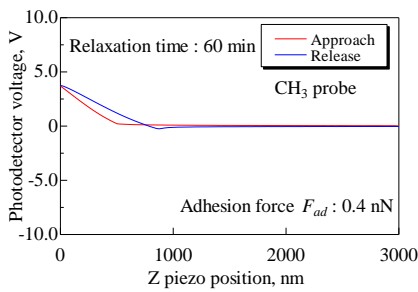

(**c**) After 60 min force curve using CH$_3$ probe

**Figure 6.** Comparison of force curves using the COOH and CH$_3$ probes without relaxation time and after 60 min. In comparison to (**a**), the adhesion force of (**b**) decreases by approximately 50%. The 6.1 nN without relaxation includes the electrostatic force, and it is thought that 3.3 nN is the adsorption force of COOH.

### 3.3. Effect of Dispersion on Force Curve by Installing and Removing the Chip

We conducted a test to investigate the effect of dispersion on force curves by installing and removing a chip. Mica was used as a substrate. Force curves were obtained under the following conditions: Sweep distance $L$ = 3000 nm, sweep speed $v$ = 600 nm/s, and cantilever load on the substrate $P$ = 4 nN (by adjusting sweep start position $L_s$). The two states of setting the chip with long and short protrusions from the holder are shown in Figure 7. We investigated the effect of the two chip states on PD sensitivity $S$ by obtaining 10 force curves and comparing the two chip states. PD sensitivities in each state are shown in Table 3. Each mean value had a difference of about 1.6 times. There are two test condition differences between the long-chip state and short-chip state as follows: (1) Protrusion length of the chip from the holder and (2) laser position on the cantilever.

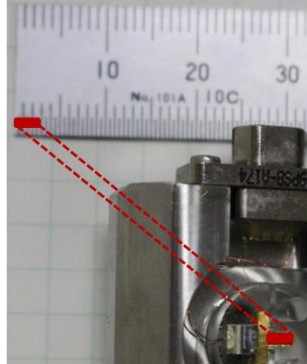 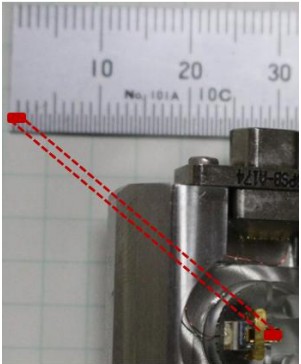

(**a**) For the long chip distance (about 3 mm)　　(**b**) For the short chip distance (about 2 mm)

**Figure 7.** Test conditions of chip distance on the cantilever holder. The figure to the left shows a cantilever installed in the holder and protruding out about 3 mm. The right shows the cantilever protruding out about 2 mm.

**Table 3.** PD sensitivity of long and short chip distance.

| Test | PD Sensitivity $S$, nm/V | |
|------|------|------|
| No. | Long | Short |
| 1 | 54.56 | 92.35 |
| 2 | 54.76 | 92.85 |
| 3 | 54.39 | 91.23 |
| 4 | 54.61 | 91.34 |
| 5 | 55.66 | 93.57 |
| 6 | 54.51 | 91.53 |
| 7 | 54.66 | 92.96 |
| 8 | 55.25 | 90.94 |
| 9 | 55.44 | 92.19 |
| 10 | 55.59 | 92.89 |
| Mean | 54.94 | 92.18 |

First, to explain difference (1), models of cantilever loaded uniform load by inertial resistance and viscous resistance are shown in Figure 8. The cantilever deflection $\delta$ at point $x = l$ is given by

$$\delta = \frac{wl^4}{8EI} \tag{4}$$

where $w$ is the summation of the uniform load of inertial resistance and viscous resistance (= $w_1 + w_2$), $E$ is Young's modulus, and $I$ is the second moment of area. When the uniform load of inertial resistance is $w_1$ and viscous resistance is $w_2$, these are given by

$$w_1 = C_D \frac{\rho}{2} v^2 a \tag{5}$$

$$w_2 = \frac{\mu v a}{l} \tag{6}$$

where $C_D$ is the drag coefficient of the cantilever, $\rho$ is the oil density, $v$ is the sweep speed, and $\mu$ is the viscosity of oil. When the values in this test condition are $E$ = 260 GPa [19] (Young's modulus for $Si_3N_4$), $I$ = 0.016 mm$^4$, $C_D$ = 1.8 [20], $\rho$ = 0.834 g/cm$^3$, $v$ = 600 nm/s, and $\mu$ = 34.6×10$^{-3}$ Pa·s, the deflection of the cantilever $\delta$ is given by

$$\delta = 2.257 \times 10^{-8} \text{ nm} \tag{7}$$

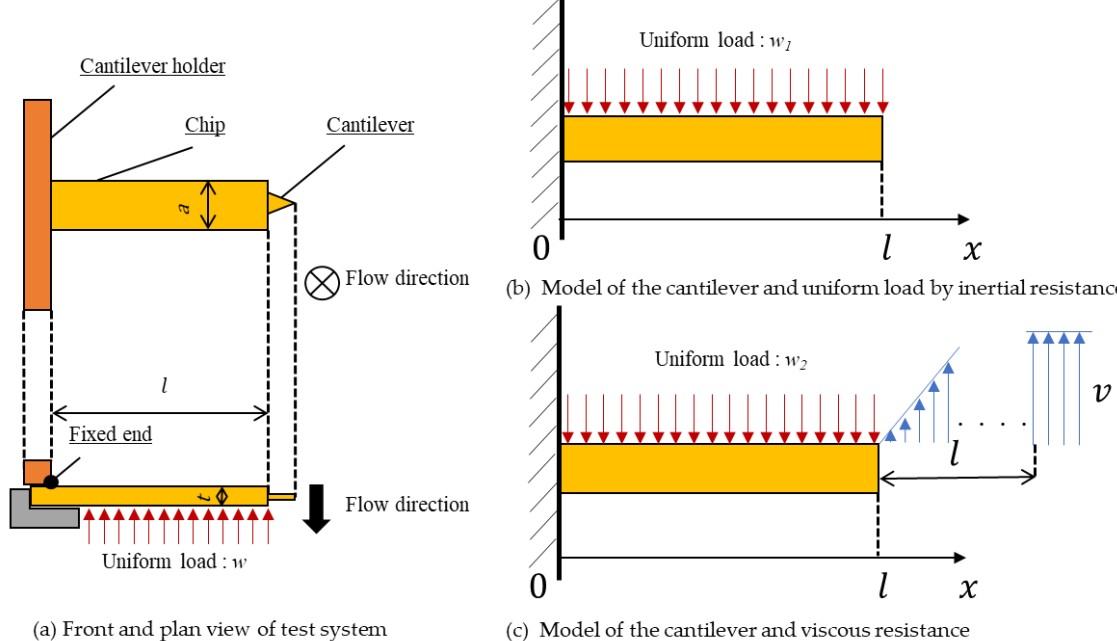

(a) Front and plan view of test system　　　　(c) Model of the cantilever and viscous resistance

**Figure 8.** Model of the test system. The effect on the cantilever from deflection by inertial resistance and viscous resistance is calculated. In (**c**), it is supposed that the relative flow velocity of oil at a distance of *l* from the cantilever is equivalent to sweep speed *v*.

It is assumed that the deformation has little effect on the dispersion of PD sensitivity, even though deformation is one of the error factors.

Second, to explain difference (2), the laser position, the relationship between the cantilever and laser position is shown in Figure 9. This test was conducted to confirm the effect on PD sensitivity of installing and removing the cantilever by changing the two chip states between long and short protrusion. However, this test includes the deviation of the laser position on the cantilever in addition to the chip states. Although the operator tried to irradiate position (i) shown in Figure 9, the laser position could not be visually checked, due to the structure of the AFM apparatus, and the position was inferred from the laser intensity and the state of intensity. For instance, there is a possibility of irradiating location (ii) shown in Figure 9. It is suggested that the deviation in the laser position greatly affects PD sensitivity. For both reasons, it was clear that the effect of deviation on the force curve by installing and removing the chip is significant.

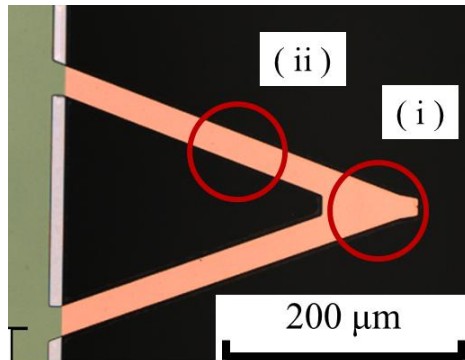

**Figure 9.** Positional relation of cantilever and laser, as imaged by a laser microscope. The operator of the AFM is required to irradiate location (**i**). However, it is impossible to visually check the laser position due to the structure of the AFM. Therefore, the operator might irradiate location (**ii**), and incorrect laser position affects the PD sensitivity.

### 3.4. Effect of Elastic Deformation of Substrate on the Force Curve

We conducted an additional test to examine the effect of elastic deformation of a substrate on the force curve. Mica and silicon wafers were used as substrates. The load of the cantilever on substrate $P$ could not be unified, because of the dispersion of PD sensitivity. Therefore, load distance $L_l$ was defined as shown in Figure 10 and the test condition was unified by load distance. Force curves were obtained under the following conditions: Sweep distance $L$ = 3000 nm; sweep speed $v$ = 600 nm/s; and load distances $L_l$ = 1250, 1000, 750, 500, and 250 nm. Force curves were obtained 10 times for each load distance. The force curves with the smallest difference between the load distance for mica and silicon wafers are shown in Figure 11. The mean PD sensitivity at each load distance is shown in Table 4.

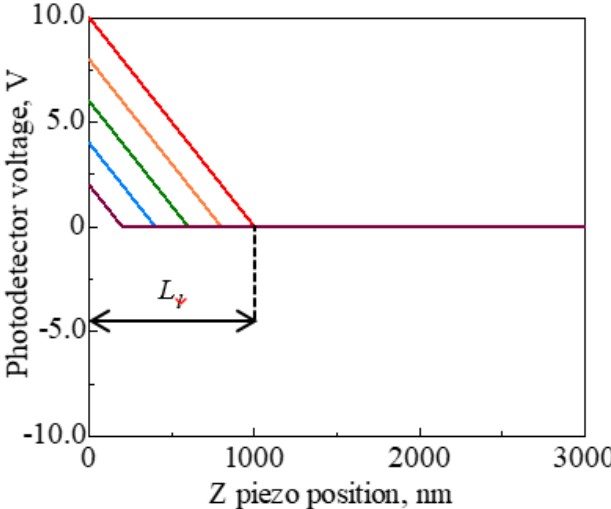

**Figure 10.** Definition of load distance $L_l$ on the force curve. The load of the cantilever on substrate $P$ cannot be unified, due to the dispersion of PD sensitivity. Thus, the test condition is unified by load distance.

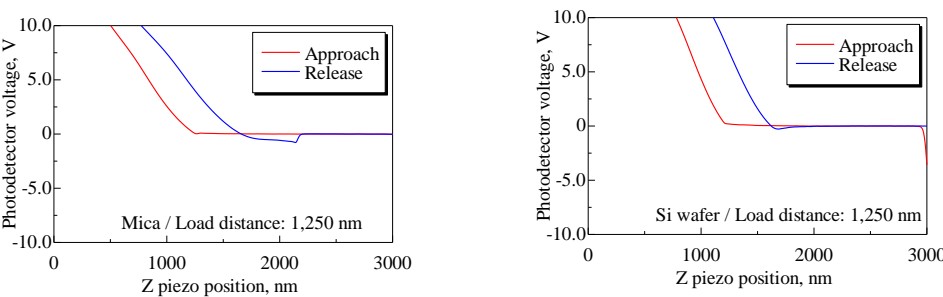

(**a**) Force curves for mica/silicon wafer in load distance: 1250 nm

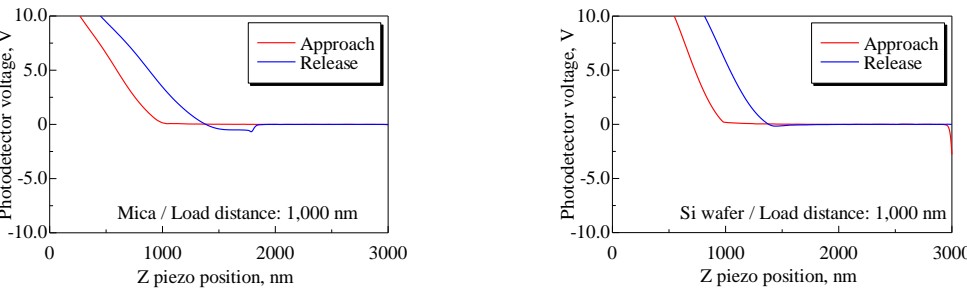

(**b**) Force curves for mica/silicon wafer in load distance: 1000 nm

**Figure 11.** *Cont.*

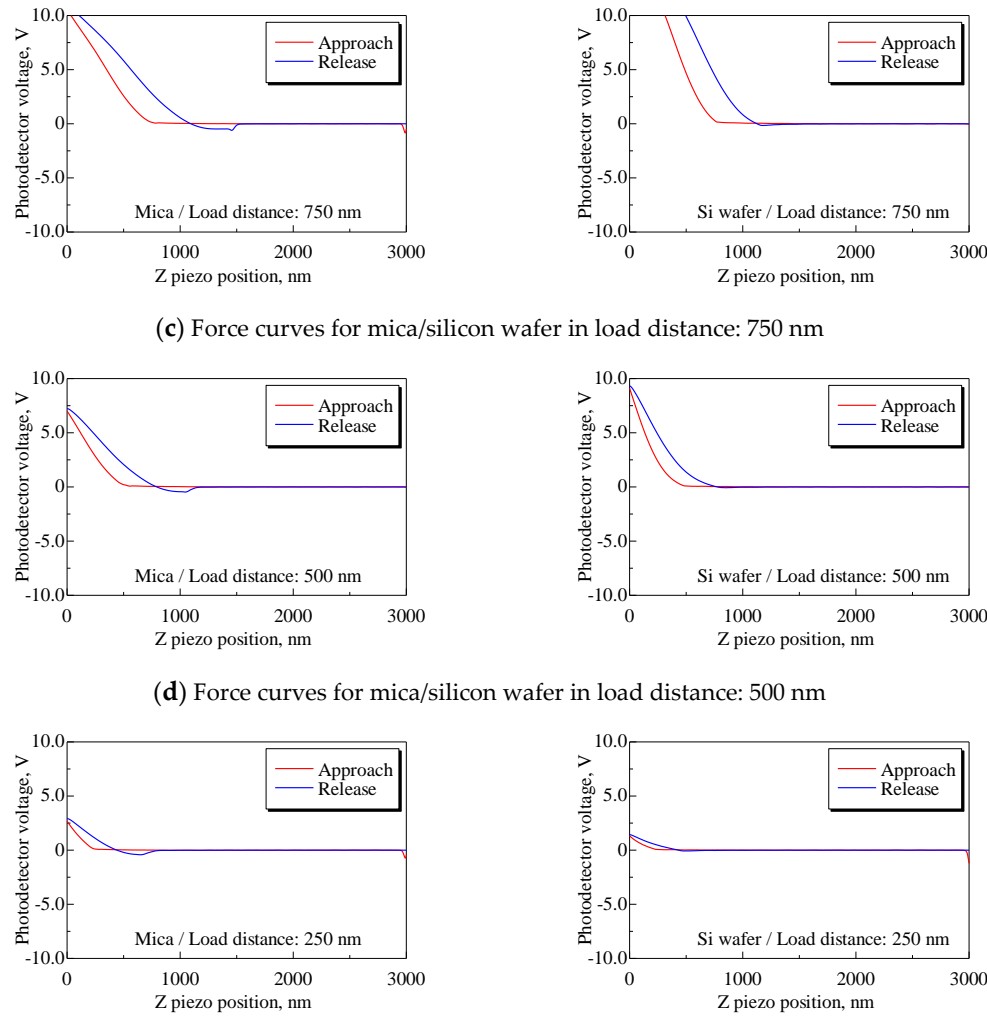

(**c**) Force curves for mica/silicon wafer in load distance: 750 nm

(**d**) Force curves for mica/silicon wafer in load distance: 500 nm

(**e**) Force curves for mica/silicon wafer in load distance: 250 nm

**Figure 11.** Force curves for mica and silicon wafers for each load distance. Both slopes of the approach process, especially at load distances $L_l$ = 1250, 1000, and 750 nm, have differences, and the mean PD sensitivities for mica and silicon have an approximately twofold difference, as shown in Table 4.

**Table 4.** PD sensitivity of long and short chip distance.

| Load distance $L_l$, nm | Mean PD Sensitivity $S$, nm/V | |
| --- | --- | --- |
| | Mica | Silicon wafer |
| 1250 | 68.71 | 39.35 |
| 1000 | 61.80 | 36.44 |
| 750 | 72.37 | 36.64 |
| 500 | 72.46 | 163.15 |
| 250 | 130.38 | 264.97 |

At load distances $L_l$ = 1250, 1000, and 750 nm, the mean PD sensitivity for mica and silicon had about a twofold difference. The models of the system are shown in Figure 12 to explain this mechanism. Figure 12a shows the force curve for mica at a load distance of 1250 nm and the model of the AFM measurement system, and Figure 12b shows that for the silicon wafer at the same load distance. First, (a) represents the model for mica, which assumes that the cantilever deforms while the mica deforms and contacts the substrate, as shown in sketch (a). In this case, the cantilever deformation changes the photodetector voltage in a positive direction and the deformation of mica changes the voltage in a negative direction. As a result, it is suggested that the two phenomena balance and make

the slope small, that is, the PD sensitivity is large. By contrast, Figure 12b is a model for a silicon wafer that assumes that the cantilever only deforms because Young's modulus of silicon is larger than that of mica, as shown in Table 5. Therefore, it appears that PD sensitivity depends only on cantilever deformation, which means that the PD sensitivity for silicon is more accurate than that for mica. Meanwhile, at load distances $L_l$ = 500 and 250 nm, both PD sensitivity values increase. The selectable range of detecting force area is limited according to the shortened load distance. Thus, the data that include adhesion energy are used to convert PD sensitivity, and that is the cause of increased PD sensitivity [21].

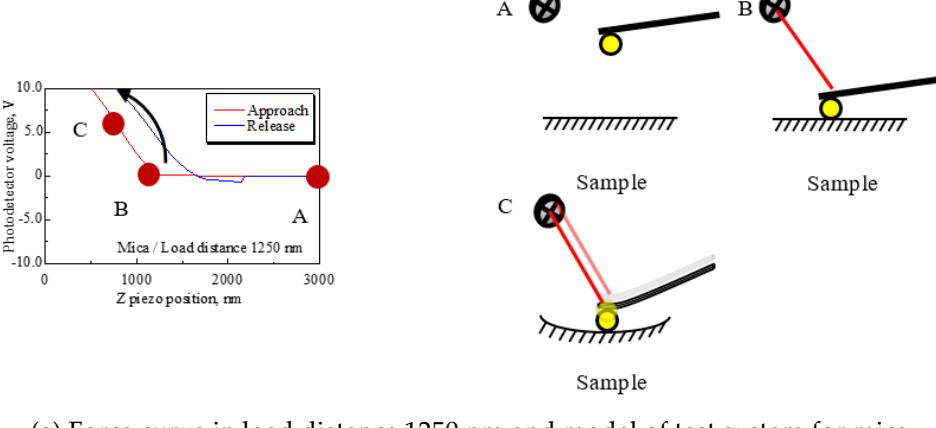

(**a**) Force curve in load distance 1250 nm and model of test system for mica

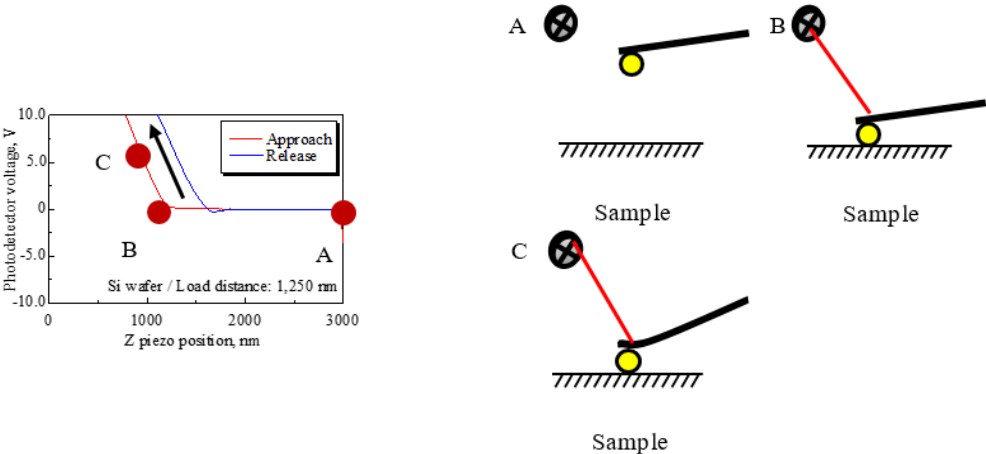

(**b**) Force curve in load distance 1250 nm and model of test system for silicon wafer

**Figure 12.** Mechanism of the PD sensitivity difference between mica and silicon. In the force curve of (**a**), the slope of the approach process—namely, PD sensitivity—changes due to deformation of the mica surface. However, the slopes for silicon do not change, because the silicon surface does not deform. That is why it is thought that both PD sensitivities have differences, and the PD sensitivity for silicon is more appropriate than mica for converting forces.

**Table 5.** Properties of mica and silicon wafer.

|  | Young's Modulus, GPa | Poisson's Ratio |
| --- | --- | --- |
| Mica [1] | 34.5 | 0.205 |
| Silicon wafer [2] (Silicon monocrystal (110)—plane) | 170 | 0.289 |

[1] Measured value [22], [2] Theoretical value [23].

### 3.5. Normality of PD Sensitivity Difference Obtained by an Operator

The error distribution needs to have normality to demonstrate that the PD sensitivity difference discussed in Section 3.3 is a random error. One test was defined as installing a cantilever in the holder, obtaining a force curve and PD sensitivity, and then removing the cantilever from the holder. To examine the normality, we conducted 400 tests and processed the data as histograms. As mentioned in Section 3.4, because silicon is more accurate than mica for obtaining PD sensitivity, the silicon wafer was used as a substrate. Force curves were obtained under the following conditions: Sweep distance $L = 3000$ nm, sweep speed $v = 600$ nm/s, load distance $L_l = 1000$ nm, and number of tests $N = 400$. The histogram is shown in Figure 13 and the data were processed as normal Q–Q plots [24]. The scatter diagram is shown in Figure 14. In the figure, $r$ is the correlation coefficient of a normal Q–Q plot, and as $r$ approaches 1, the distribution becomes more Gaussian. As a result, it was demonstrated that the error distribution is Gaussian.

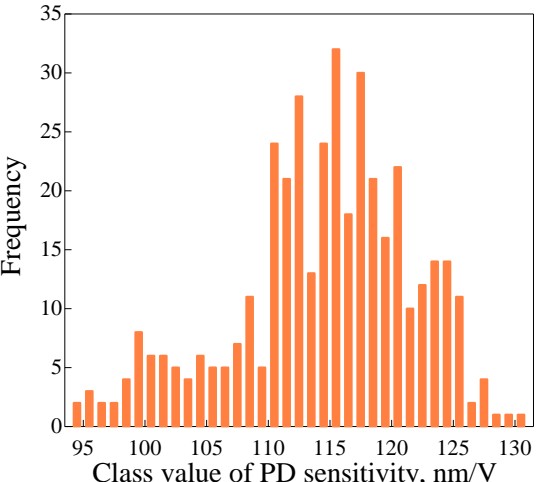

**Figure 13.** Histogram of PD sensitivity from 400 tests. PD sensitivities are obtained between the COOH-modified probe and the silicon wafer.

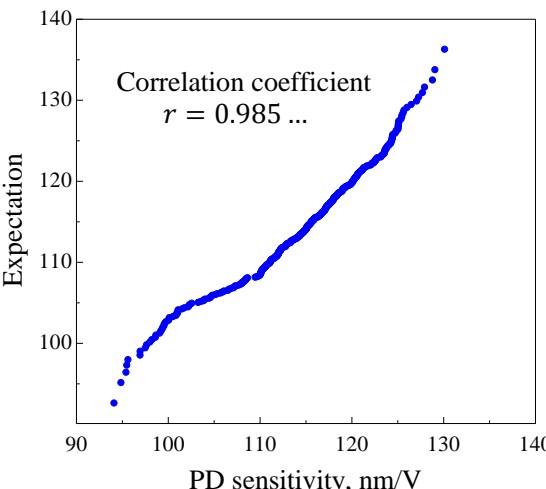

**Figure 14.** Normal Q–Q plot for PD sensitivity from 400 tests. As the correlation coefficient of sample data processed with normal Q–Q plots approaches 1, it is suggested that the distribution of the samples is Gaussian. Therefore, the distribution of PD sensitivity is considered to be a Gaussian distribution.

We propose the following method for calibrating PD sensitivity. When the population mean of PD sensitivity is $\mu$, this satisfies the inequality

$$\overline{S} - t_{(N-1,\alpha)} \sqrt{\frac{V}{N}} < \mu < \overline{S} + t_{(N-1,\alpha)} \sqrt{\frac{V}{N}} \tag{8}$$

where $\overline{S}$ is the sample mean of PD sensitivity, $\alpha$ is the significance level, $t_{(N-1,\alpha)}$ is the value of Student's t-test when the degree of freedom is $N$–1 and the significance level is $\alpha$, $V$ is unbiased variance, and $N$ is the parameter in this test. Then, the confidence interval *CI* is given by

$$CI = 2t_{(N-1,\alpha)} \sqrt{\frac{V}{N}} \tag{9}$$

By defining a permissible error, a necessary number of tests for fitting within the permissible error range, namely, theoretical sample size $n$, can be calculated with probability 1–$\alpha$ (confidence coefficient). When the one-sided permissible error is $\delta$, this satisfies the inequality

$$\delta \geq \frac{CI}{2} = t_{(N-1,\alpha)} \sqrt{\frac{V}{N}} \tag{10}$$

Then, we solve for $n$ as follows:

$$n \geq V \left( \frac{t_{(N-1,\alpha)}}{\delta} \right)^2 \tag{11}$$

When the defined value is $\delta$ = 1.5 nm/V, $n$ is given by

$$n \geq 89.590\ldots \tag{12}$$

As a result, it is revealed that the theoretical sample size in this test is about 100 tests. Thus, this calibration method of PD sensitivity by Equation (8) gives the PD sensitivity having an appropriate error range.

The above analysis allows us to propose a more accurate adsorption force measurement method. A flow chart of the method is shown in Figure 15. It should be noted that PD sensitivity ought to be calibrated after force curves are obtained for the substrate and polar group for which the adsorption force is to be measured. This sequence is intended to prevent the PD sensitivity calibration procedure from affecting the accuracy of the adsorption force measurement.

To show the verification of this calibration method, the adsorption force onto mica of both the new measurement and previous study (Reference [13]) was compared by this calibration method. Table 6 shows the adsorption force of COOH onto the mica substrate loaded on $P$ = 3 nN of both uncalibrated and calibrated. The uncalibrated new adsorption force data and the previous adsorption force data differed by 0.15 to 0.67 nN. However, the calibrated adsorption force data having some error bar were almost the same. These facts indicate that the calibration method is significant to measure and compare the adsorption force values and understand the meaning of the difference of subnano-order.

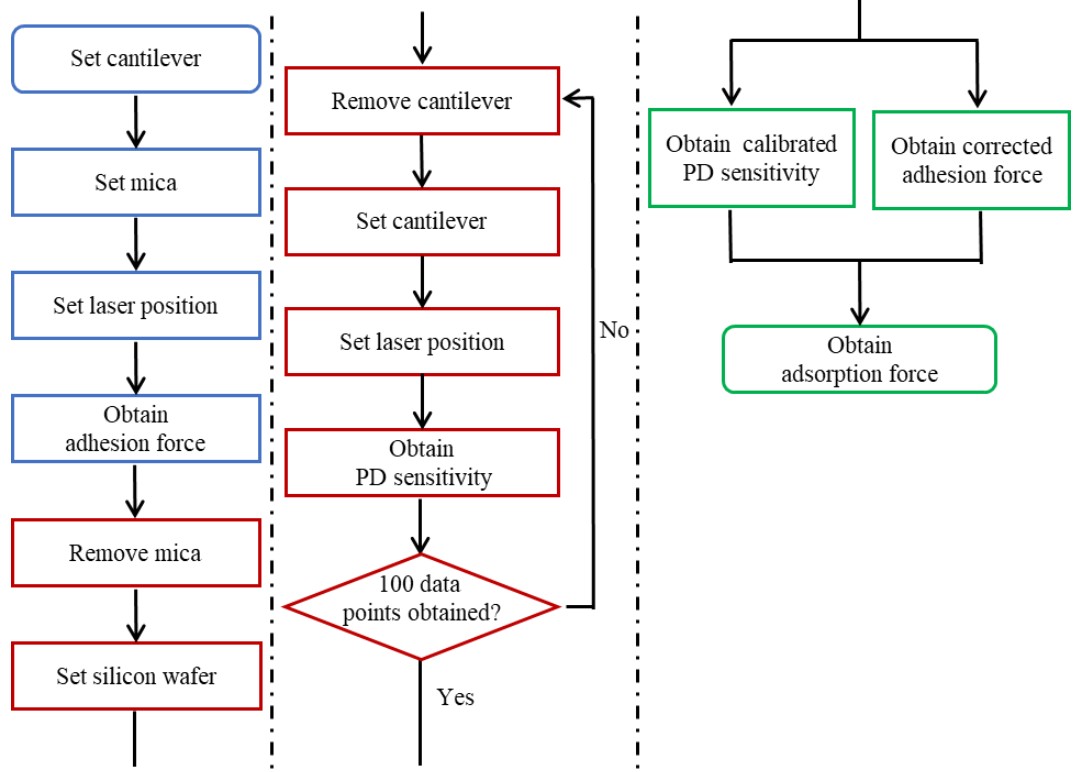

**Figure 15.** Flow chart of the calibration method. Note that PD sensitivity should be calibrated after obtaining force curves to avoid affecting the measurement accuracy. If calibration is not performed before obtaining force curves, the modified polar group might be affected.

**Table 6.** Uncalibrated and calibrated adsorption force of COOH onto mica of new measurement and Reference [13] (the probe load on $P$ = 3 nN).

| New: Uncalibrated | | New: Calibrated | | |
|---|---|---|---|---|
| No. | Adsorption force, nN | No. | Adsorption force, nN | Error bar, nN |
| 1 | 2.861 | 1 | 3.172 | 0.090 |
| 2 | 2.729 | 2 | 3.219 | 0.091 |
| 3 | 2.760 | 3 | 3.202 | 0.091 |
| 4 | 2.767 | 4 | 3.195 | 0.090 |
| 5 | 2.720 | 5 | 3.150 | 0.089 |
| **Reference [13]: Uncalibrated** | | **Reference [13]: Calibrated** | | |
| No. | Adsorption force, nN | No. | Adsorption force, nN | Error bar, nN |
| 1 | 2.570 | 1 | 3.147 | 0.012 |
| 2 | 2.437 | 2 | 3.137 | 0.012 |
| 3 | 2.369 | 3 | 3.128 | 0.012 |
| 4 | 2.266 | 4 | 3.135 | 0.012 |
| 5 | 2.191 | 5 | 3.091 | 0.012 |

## 4. Conclusions

To quantitatively and directly evaluate the adsorption force between probes modified with $CH_3$ and COOH and mica and silicon substrates, this study revealed the error factors and proposed an error analysis method.

1. As the sweep speed increases, so does the fluid resistance to a cantilever; in other words, a cantilever detects a repulsive force before contacting the substrate. To keep the non-detecting area of a force curve horizontal, it is necessary to set an appropriate sweep speed according to the spring constant of the cantilever.

2.  It was demonstrated that the effect of the electrostatic force on the force curve is eliminated by securing a substrate with conductive tape and allowing for charge relaxation for a few hours. In this study, the electrostatic force was eliminated by using carbon tape and letting the sample rest for 60 min.

3.  The installation position of a cantilever in the holder and the laser position on the cantilever are factors that alter force curves in AFM measurement. In this study, despite the same cantilever, the PD sensitivity was about 1.6 times different with different physical configurations.

4.  Two tests using mica and silicon as substrates showed that PD sensitivity is largely influenced by elastic deformation of the substrate. It was suggested that accurate PD sensitivity can be obtained for a cantilever by calibrating the PD sensitivity with a hard substrate like silicon. In this study, a silicon wafer gave an accurate PD sensitivity.

5.  The distribution of the PD sensitivity deviation (due to repeated reinstallation of the cantilever and laser positioning system) was Gaussian. We proposed a calibration method for obtaining accurate PD sensitivity. In this study, when we set the one-sided permissible error of PD sensitivity as 1.5 nm/V, the theoretical sample size was approximately 100 tests.

**Author Contributions:** Conceptualization, T.H.; Data curation, W.Y.; Investigation, W.Y.; Methodology, W.Y.; Resources, T.H., K.T. and K.N.; Supervision, T.H.; Writing – original draft, W.Y.; Writing – review and editing, W.Y. and T.H. All authors have read and agreed to the published version of the manuscript.

**Funding:** This research received no external funding.

**Conflicts of Interest:** The authors declare no conflict of interest.

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
