# Peer review of "Study on the Quantitative Evaluation of the Surface Force Using a Scanning Probe Microscope"

_lubricants, doi:10.3390/lubricants8060066_

Round 1
Reviewer 1 Report
The author used a lot of testing to certify the adhension force and setup the method. The results seemed to fill in the scope of Lubricants. However, I have several questions:
1. How groups of CH3 and COOH were determined? or the detailed information of modification should be provided.
2. How many samples were used for Rq measurement as well as other measurements?
3. Line 136 first word should be deleted.
4. Line 32 " reacting chemically.." should chemically reacting
5. How to consider the surface roughness difference between mica and silicon on the adhension force?
6. Was the probe worn after hundred sweeping?
7. Line 38 need several literature supporting, e.g Soltanahmadi S,. Tribology Letters, 2019, 67(3): 80. and Dong G,. Tribology International. 2018, 127:302-312
Reviewer 2 Report
The topic of the paper is VERY INTERESTING. The paper is also written and organized well. However, a few suggestions to further improve the paper are given below.
- The last statement in the Abstract is confusing. I guess there is some error in it. Please check and rectify.
- Section 2.4: The word Paraffinic at the start of the paragraph is misplaced, I think.
- The rest of the paper is written well with the results presented in a logical order.
- The authors in their previous study Ref-11, did develop the method of measuring the adsorption forces. However, in the present paper they have developed a more accurate and a sensitive methodology by including the statistical analysis. So, I strongly suggest that a small paragraph at the end of the manuscript should be added to highlight the significance of the present study in light of the study that they conducted in Ref-11.
- I suggest to have a comparison of the force values obtained from their previous study and the present study under similar conditions would greatly help the readers to understand the significance of this study.
Round 2
Reviewer 1 Report
The revised version now has met the journal standard. The author has addressed all the concerns.